# Mercury-methylating bacteria are associated with copepods: A proof-of-principle survey in the Baltic Sea

**Elena Gorokhova**[1]***, **Anne L. Soerensen**[1¤], **Nisha H. Motwani**[2]

**1** Department of Environmental Science, Stockholm University, Stockholm, Sweden, **2** School of Natural Sciences, Technology and Environmental Studies, Södertörn University, Huddinge, Sweden

¤ Current address: Department of Environmental Research and Monitoring, Swedish Museum of Natural History, Stockholm, Sweden
* elena.gorokhova@aces.su.se

**Data Availability Statement:** All relevant data are within the paper and its Supporting Information files.

## Abstract

Methylmercury (MeHg) is a potent neurotoxin that biomagnifies in marine food webs. Inorganic mercury (Hg) methylation is conducted by heterotrophic bacteria inhabiting sediment or settling detritus, but endogenous methylation by the gut microbiome of animals in the lower food webs is another possible source. We examined the occurrence of the bacterial gene (*hgcA*), required for Hg methylation, in the guts of dominant zooplankters in the Northern Baltic Sea. A qPCR assay targeting the *hgcA* sequence in three main clades (Deltaproteobacteria, Firmicutes and Archaea) was used in the field-collected specimens of copepods (*Acartia bifilosa*, *Eurytemora affinis*, *Pseudocalanus acuspes* and *Limnocalanus macrurus*) and cladocerans (*Bosmina coregoni maritima* and *Cercopagis pengoi*). All copepods were found to carry *hgcA* genes in their gut microbiome, whereas no amplification was recorded in the cladocerans. In the copepods, *hgcA* genes belonging to only Deltaproteobacteria and Firmicutes were detected. These findings suggest a possibility that endogenous Hg methylation occurs in zooplankton and may contribute to seasonal, spatial and vertical MeHg variability in the water column and food webs. Additional molecular and metagenomics studies are needed to identify bacteria carrying *hgcA* genes and improve their quantification in microbiota.

## Introduction

Mercury (Hg) is a global pollutant adversely affecting human and wildlife health due to its toxicity and distribution in the environment [1]. Various processes, both natural and anthropogenic, lead to the release of primarily inorganic Hg (IHg), which can undergo methylation resulting in formation of neurotoxic monomethylmercury (MeHg). While both IHg and MeHg can be taken up by biota, only MeHg bioaccumulates in aquatic food webs [1,2].

The primary pathway for MeHg production is microbial Hg methylation [3], and a bacterial gene cluster associated with such methylation (*hgcAB*) has recently been discovered [4,5]. It was previously thought that mainly sulfate-(SRB) and iron-(FeRB) reducing bacteria methylate

**Funding:** Funding from The Swedish Research Council (Vetenskapsrådet, grant number 2018-05213) and the Swedish Research Council for Environment, Agricultural Sciences and Spatial Planning (FORMAS) to EG (grant number 2018-01010) and ALS (grant number 2016-00875) is acknowledged. The funders had no role in governing the research or data usage.

**Competing interests:** The authors have declared that no competing interests exist.

Hg in anoxic conditions [6–8]. However, the *hgcAB* gene cluster has been identified in some syntrophic and fermentative *Firmicutes* indicating a broader phylogenetic and functional representation of Hg methylators [9]. Recently, clade-specific quantitative PCR (qPCR) assays were developed to quantify the abundance of *hgcA* gene of the main methylators [10]. Hence, *hgcAB* and *hgcA* distribution can be used to predict occurrence of potential Hg methylators in the environment [11]. Understanding *hgcAB* and *hgcA* distribution is essential for estimating MeHg production in the water column and biomagnification in food webs [12].

Worldwide, great differences in MeHg accumulation have been reported for similarly structured and geographically close food webs [2,12]. In aquatic environments, MeHg production takes place in both sediment and water column [12,13]; however, in the oxygenated waters, Hg methylation may occur in anoxic microenvironments on sinking organic matter [7]. In the water column, MeHg, bioconcentrated by phytoplankton, heterotrophic biofilms and periphyton, enters the food web via zooplankton grazing, with subsequent transfer of zooplankton-associated MeHg to zooplanktivores [12,14,15]. An additional source of MeHg and a possible contributor to the variability in food-web bioaccumulation could be endogenous Hg methylation by gastrointestinal microbiota [5,16] with subsequent MeHg uptake by the host. Therefore, endogenous Hg methylation in primary consumers could constitute an unexplored MeHg source with consequences for higher trophic levels. Exploring the Hg methylation capacity of gut microbiota has been attempted in various animals using both analytical and molecular approaches [16]. While the gene cluster *hgcAB* has been identified in the gut microbiome of some terrestrial arthropods [5,16], its status in aquatic invertebrates is so far unknown.

In the Baltic Sea, Hg sources are historically high, due to both natural and anthropogenic inputs [17], which should promote Hg methylation ability in microorganisms [3] and facilitate establishment of methylators in microbiota of filter-feeders, such as zooplankton. The objective of our study was to conduct a field survey to identify whether the *hgcA* gene is present in the gut microbiome of zooplankton in the Baltic Sea. Our findings reported here represent the first record of potential methylators associated with zooplankton and imply that endogenous Hg methylation might occur in primary consumers as a pathway by which MeHg can enter the food webs.

## Materials and methods

### Ethics statement

The sampling was conducted within Swedish and Finnish Marine Monitoring Programmes in the Baltic Sea and SYVAB's marine monitoring program in the Himmerfjärden Bay (Himmerfjärden Eutrophication Study; www2.ecology.su.se), and no specific permissions were required for any of the sampling locations in this study. Also, we did not require an ethical approval to conduct this study as no animals considered in any animal welfare regulations and no endangered or protected species were involved.

### Field zooplankton collections and sample preparation

We focused our survey on microcrustaceans, cladocerans and copepods, which are the major groups of mesozooplankton in the Baltic Sea. These microscopic animals are largely herbivorous, with parthenogenic cladocerans thriving in the mixing layer and reproducing mostly during summer, whereas copepods usually reside at deeper layers performing vertical migrations related to onthogeny, temperature and predation risk [18].

Zooplankton were collected at four stations in the coastal and open sea area of the northern Baltic Proper and the Bothnian Sea (Table 1, S1 Text, S1 Fig). Samples were taken by vertical

**Table 1. Summary of zooplankton samples used for qPCR analysis.** Species abbreviations for copepods: *Acartia bifilosa* (Ab, adults), *Eurytemora affinis* (Ea, adults), *Limnocalanus macrurus* (Lm, CIV), and *Pseudocalanus acuspes* (Pa, CIV), and cladocerans: *Bosmina coregoni maritima* (Bm, body length > 0.7 mm) and *Cercopagis pengoi* (Cp, > 2mm, excluding the tail spine). In total, 33 field-collected zooplankton samples and 3 reference samples (*Artemia* spp.) were analyzed.

| Station | Location, area | Geographic coordinates and bottom depth | Month, Year | Sampling depth, m | Number of samples per species | | | | | |
|---|---|---|---|---|---|---|---|---|---|---|
| | | | | | Ab | Ea | Lm | Pa | Bm | Cp |
| H4 | Himmerfjärden Bay, Northern Baltic Proper, Swedish coast | N 58˚59', E 17˚43'; 30 m | Jun 2007 | 28–0 | | 3 | | | 3 | 2 |
| BY31 | Landsort Deep, Northern Baltic Proper, open sea | 58˚35' N, 18˚14' E; 454 m | Jun 2009 | 100–60 | | | | 3 | | |
| | | | | 30–0 | 3 | 3 | | | 3 | |
| F64 | Åland Sea, open sea | N 60˚11', E 19˚08'; 285 m | Sep 2009 | 100–0 | 3 | | | | | |
| US5b | Bothnian Sea, open sea | N 62˚35', E 19˚58'; 214 m | Aug 2006 | 100–0 | 3 | | 4 | 3 | | |

tows at 0.5 m/s with a WP2 net (mesh size 90 or 100 μm; ring diameter 57 cm) equipped with a cod end. At some stations, bottom to surface tows were taken, and at others, we used either stratified tows or sampled only an upper part of the water column.

Animals retrieved from the cod-end were placed in 0.2-μm filtered aerated seawater and supplied with an excess of the cryptophyte *Rhodomonas salina* (strain CCAP 978/24) to clear the guts of any potential *hgcA*-containing microorganisms associated with their food items and only retain those microbes closely associated with the gut mucosa. The animals were transferred to the new medium containing the fresh algal suspension two-three times. This procedure was applied to all species except *Cercopagis pengoi*, a predatory onychopod, feeding by puncturing exoskeleton of planktonic crustaceans and sucking soft body tissues [19]. Such feeding mode leaves the chitinous gut of the prey intact in the discarded carcass, hence, the contamination of the predator gut with prey microflora was considered unlikely, and *C. pengoi* were not subjected to the gut clearance procedure. For the rest of the zooplankton, randomly selected individuals with visibly reddish guts (indicating that the animals were active and feeding during the incubation) were selected following two-hour incubation. All specimens were preserved in groups using RNA*later* and stored at –20˚C [20].

From the RNA*later*-preserved samples, different species of copepods and cladocerans were picked under a dissecting microscope with forceps, rinsed in artificial seawater, and transferred in groups (30–50 ind. sample[-1]) into Eppendorf tubes. The following species and developmental stages were selected for the analysis: (1) copepodites (CV–VI) of *Acartia bifilosa* and *Eurytemora affinis*; these are small calanoids, dominant in the study area and present all year round, mostly in the epipelagia; (2) copepodites (CIII-IV) of *Limnocalanus macrurus* and *Pseudocalanus acuspes*; these are large calanoid copepods, dominant zooplankton below the halocline in the Northern Baltic, and important prey for zooplanktivores; (3) cladoceran *Bosmina coregoni maritima* (females, >0.7 mm); a small zooplankter, often reaching high abundance in the surface waters during summer and being occasionally important prey for zooplanktivorous fish, and (4) cladoceran *Cercopagis pengoi* (Barb Stages II and III); a large predatory zooplankter representing a secondary consumers a common prey for fish during summer. Thus, except for *C. pengoi*, all analyzed species are primary consumers and dominant species in the pelagic food web.

Reference samples used as a contamination control were hatched *Artemia* spp. nauplii (San Francisco Bay Brand) grown on axenic culture of *R. salina* ($5 \times 10^4$ cells mL[-1]) in artificial seawater (28 g L[-1] of Instant Ocean synthetic sea salt; Aquarium Systems Inc., Sarrebourg, France). The animals were sacrificed after reaching a body length of ~2 mm and treated in the same way as the zooplankton samples. As no amplification was ever observed in the reference samples with *Artemia* guts (3 replicates, 25 guts sample[-1]), we beleive that no false positives

were produced, and bacterial contamination during experimental procedure and sample preparation was either negligible or non-existent.

## DNA extraction

From each specimen, the gut was excised with a sharp needle, a pair of ultrafine forceps under dissecting microscope; the instrumentation and glassware were sterile. In total, 36 samples, 25–50 guts sample$^{-1}$ (depending on the animal size), were prepared (Table 1, S1 Text). The guts were transferred into 1.5 mL centrifuge tubes for Chelex-based DNA extraction [21] following a protocol developed for analysis of prokaryotes in zooplankton [22]; See S1 Text for details on the laboratory procedures and S1 Table for the DNA yield in different species (Supporting Information).

## qPCR assay

Three main clades were considered as potential *hgcA*-targets, Deltaproteobacteria, Firmicutes, and Archaea, the Hg-methylators broadly present in the environment that have been reported to carry this gene [9]. For each clade, a separate SYBR Green qPCR assay was performed using a clade-specific protocol of Christensen and co-workers [10]. As a standard, a synthetic DNA oligonucleotide [23] comprising the clade-specific target sequence was constructed using a representative strain: *Dv. desulfuricans*, *Df. metallireducens*, and *Ml. hollandica*, for Deltaproteobacteria, Firmicutes, and Archaea, respectively (S2 and S3 Tables). The standards were cloned into plasmids and applied in five-step tenfold serial dilutions, $1.5 \times 10^6$ to $1.5 \times 10^2$ apparent copies of target DNA per reaction (S4 Table, S2 Fig). The qPCR primers and amplification conditions [10] were used for all test samples, reference samples, NTC and standards (S3 and S4 Tables). Under these conditions, qPCR yielded a single product in each standard and in the test samples within an assay as evidenced by the melt curve analysis (S3 Fig). No product was produced in the reference samples and NTC (non-template control) within the assay range (30 cycles).

## Data analysis

The number of *hgcA* copies detected by qPCR was used to calculate the number of *hgcA* copies per individual and per µg of zooplankter wet weight (i.e., weight-specific number of Hg methylators). In these calculations, individual zooplankter weights [24] were used (S6 Table), and one copy of *hgcA* per cell as determined by Christensen and co-workers [10]. Given substantial variations in the amplification efficiency and detection limits for these qPCR assays among different bacterial strains that have been evaluated during the method development (efficiency: 60 to 90%, detection limits: $10^2$ to $10^6$ *hgcA* copies; see the method description [10]), any statistical comparisons between species/sites were not meaningful [25]. Therefore, we consider our results largely descriptive, indicative of the presence/absence of *hgcA* and, to a lesser extent, of the interspecific or geographical variation.

## Results and discussion

All four copepod species tested positive for *hgcA* genes (Fig 1), whereas no amplification was observed for the two cladocerans. Among the clades tested, the *hgcA* genes of only Deltaproteobacteria and Firmicutes, but not Archaea, were found in the copepod guts. Although there was a substantial imbalance in the sampling effort between copepods and cladocerans (25 vs. 8 samples; Table 1), the occurrence of *hgcA*-positive samples for copepods only is suggestive of a difference. However, given the differences in the limit of quantification among the clades (S4

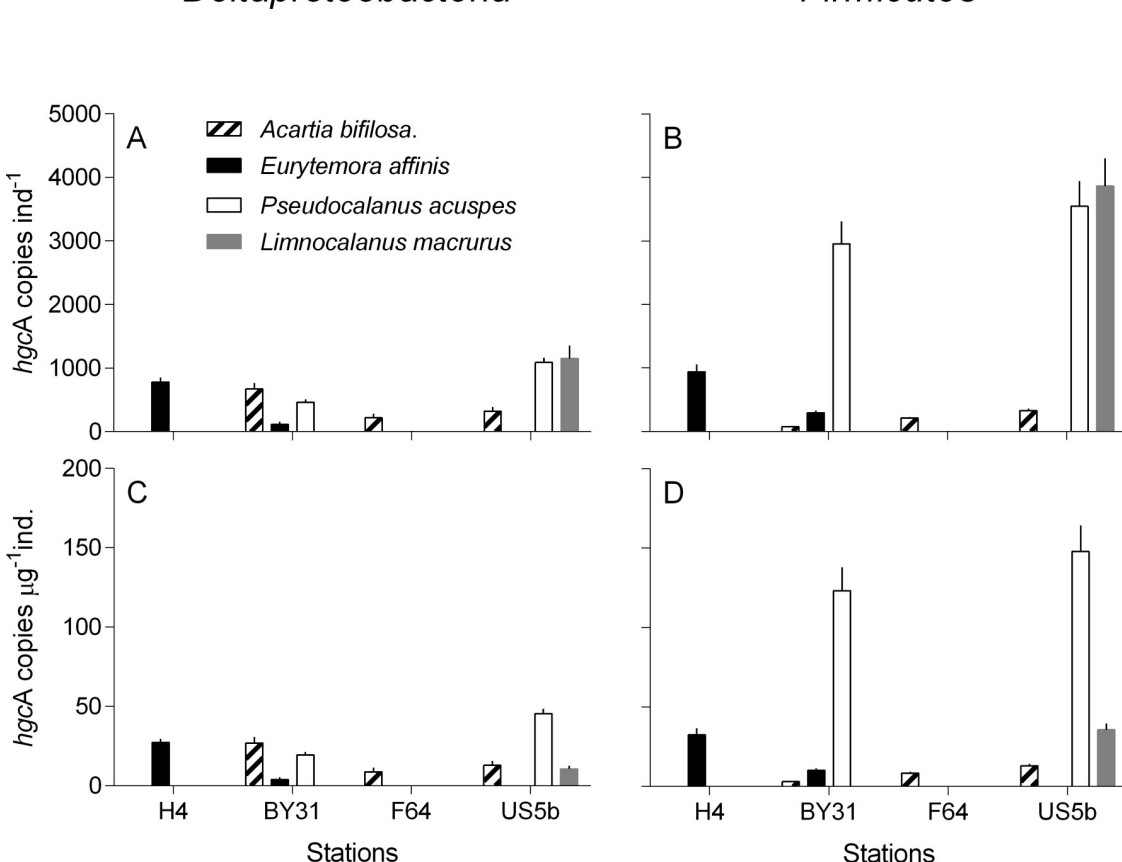

**Fig 1. Abundance of *hgcA* gene in the copepods collected in different areas of the Baltic Sea, ordered south to north.** Data are shown as mean ± SD; *n* = 3 in all cases, except *Limnocalanus macrurus*, where *n* = 4 (note that each replicate sample is composed by 25–20 dissected guts); see Table 1 for the number of replicates and S1 Fig for the map of the sampling sites. No amplification was observed in any of the cladoceran samples. The individual-specific abundance (number of *hgcA* copies per individual) is shown in the upper panels and the weight-specific abundance (number of *hgcA* copies per μg wet weight of zooplankter) is shown in the lower panels (A and C: Deltaproteobacteria and B and D: Firmicutes; no amplification was observed for Archaea). Observe that set of species is unique for every station; when no value is given, no samples for the particular species was available for the analysis.

Table), the between-clade differences in the *hgcA* abundance should be treated with caution. Moreover, although Archaea are commonly reported to occur in zooplankton guts [26], the contribution of this group can be low compared to bacteria [27]. Therefore, the lack of *hgcA* amplification in the assays with archaeal primers may–at least in part–be related to the poor representation of these microorganisms in the guts. Considering the reported variability in the amplification efficiency of the qPCR assays among different bacteria tested with these assays [10] and unknown composition of the *hgcA*-positive microbiota, only rough interpopulation comparisons are possible. However, the overall findings suggest that microbiota of zooplankters carries *hgcA* genes and thus may be capable of Hg methylation, which remains to be demonstrated. The next step is to conduct experimental studies measuring the Hg methylation capacity in the copepods and cladocerans that were found to have such striking differences in the *hgcA* microbiota.

Whether bacteria-driven Hg methylation in zooplankton guts takes place depends not only on the occurrence of *hgcA*-carrying bacteria but also on the functional performance of these bacteria. To assess the *hgcA* expression, an analytical effort using available molecular tools,

such as RT-qPCR, RNA sequencing, and RNA-SIP is required. Furthermore, a better understanding of community structure is needed. Although our results do not provide any taxonomic identification of the bacteria involved, the observed prevalence of Firmicutes among the *hgcA*-carriers (Fig 1) agrees well with a relatively high abundance of this bacterial group in the microbiome of other copepods [28,29]. In future studies, a 16S rRNA gene diversity profiling and *hgcAB* amplification with high-throughput sequencing should be combined with *hgcA* quantification [11]. Broad-scale zooplankton sampling, including seasonal, spatial and vertical coverage, should provide material for such an evaluation.

If gut Hg methylation occurs, zooplankton may serve as a primary MeHg entrance point of global significance and affect variability in MeHg transfer to secondary consumers [3]. A mass-balance budget for the herbivourous marine copepod *Calanus hyperboreus* suggested that endogenous Hg methylation could account for up to 70% of the annual MeHg uptake in this species [30]. If these estimates are correct, they might explain why reported drivers of MeHg variability are often contradictory. Indeed, MeHg concentrations in herbivorous zooplankton vary among taxa [31,32], demographic population structure [33] and growth stoichiometry [34]. In wild populations, however, these factors are difficult to disentangle [35], partly due to their ultimate dependence on body size. Todorova and co-workers [35] speculated that higher bioaccumulation of MeHg in larger species resulted from higher filtration efficiency being a function of body size, whereas Kainz and co-workers [31] attributed this size dependence to large zooplankton having larger anaerobic intestinal niches, where Hg methylation can take place [36]. Supporting the view of Kainz and co-workers [31], we found that larger copepods carried a greater number of *hgcA* copies, both per individual and per body mass. However, no amplification was observed in equally large cladoceran *Cercopagis*; the latter implies that not only body size, but also phylogenetic differences between the hosts are important. In the large-bodied copepods *L. macrurus* and *P. acuspes*, our estimate of *hgcA* genes yielded up to 10-fold higher values compared to the small-bodied *A. bifilosa* and *E. affinis*, with the difference being most pronounced for *Firmicutes* (Fig 1). The group-specific variability may affect spatial and seasonal contribution of endogenous MeHg to secondary consumers, because different zooplankton groups that vary in their ability to methylate Hg would have different capacity to contribute MeHg to bulk zooplankton. For example, at least in the Baltic Sea, the relative importance of gut Hg methylation and MeHg uptake by zooplankton would increase in winter due to the higher contribution of copepods to bulk zooplankton biomass [37].

The gut of copepods is likely to have anoxic conditions, at least in some species [36] and, thus, provides a suitable habitat for methylating microbes. Notably, the morphology of cladoceran gut predisposes it to active oxygenation, and gut microbiota in these animals is dominated by clones affiliated to aerobic or facultative anaerobic bacteria [38], which may explain the lack of *hgcA* amplification in our cladoceran samples. Hg-methylating genes have been detected in invertebrate microbiota, including termites, beetles, and oligochaetes [5,16], and in some invertebrates the endogenous MeHg production has been documented [39]. As a life form, intestinal microbiota exists in biofilms, and such communities are increasingly recognized as important sites for environmental Hg methylation [40,41]. Commensal biofilms are present in both planktonic and benthic animals that actively exchange gut and body-surface microbiota with the ambient microbial communities and other animals [42]. We found no *hgcA* genes in the gut of the predatory *Cercopagis pengoi*, which may indicate that the digestive system of predators with this feeding mode (puncturing exoskeleton of planktonic crustaceans and sucking soft body tissues) is less likely to become populated by Hg-methylating bacteria compared to filter-feeders that have a more active exchange with diverse microbial communities of seston.

The presence of Hg-methylating bacteria in copepod guts and, hence, in their carcasses and fecal pellets, could be an important and yet unquantified source for MeHg production in the water column [43]. Remineralization of organic matter is associated with elevated MeHg production [43,44], and Hg methylation potential is higher in fresh organic matter than in decomposed material [7,44]. Zooplankton fecal pellets, a considerable fraction of settling marine organic matter, are almost completely remineralized in the water column, while degraded phytoplankton and terrestrial organic matter aggregates are more likely to reach the sea floor [45]. The presence of active Hg-methylating bacteria in fecal pellets could increase Hg methylation efficiency compared to non-fecal organic matter, where a lag phase related to colonization time is expected. In the latter case, the ecological niche for Hg-methylating bacteria might not become available until the most labile parts of the organic matter are already remineralized, resulting in lower MeHg production. Ingestion of fecal pellets by mesopelagic zooplankters and benthic animals could also facilitate spread of methylators among invertebrates and enrich these consumers with microflora of epipelagic zooplankters. In addition, these pellets can become enriched in Hg methylators during the time spent in the water column. In line with this, we found higher *hgcA* abundances in *P. acuspes* and *L. macrurus* residing in deeper water layers compared with *A. bifilosa* and *E. affinis* inhabiting the epipelagic zone (Fig 1).

One can speculate that endogenous Hg methylation in zooplankton could help explain spatial and temporal trends of fish MeHg concentrations in the Baltic Sea. The strong decrease in Hg inputs to the Baltic Sea during the last decades has not resulted in a consistent decrease in fish Hg levels across the sea [17,46]. During this time, significant and basin-specific changes occurred in zooplankton communities [47] in concert with alterations in climate, nutrient inputs and terrestrial runoff [17,44]. It is plausible that synchronous shifts in the methylation capacity of zooplankton, at both the individual microbiome and community levels, have taken place contributing to the MeHg dynamics in the food web. Experimental studies and quantitative analysis of the interactions between biotic and abiotic processes governing endogenous MeHg production is therefore essential, if we are to understand uptake and bioaccumulation of MeHg in water column and food webs.

## Supporting information

**S1 Text. Laboratory procedures.**
(PDF)

**S1 Table. DNA yield for zooplankton gut samples.**
(PDF)

**S2 Table. Synthetic oligonucleotides used as standards for *hgcA* gene amplification.**
(PDF)

**S3 Table. Summary of the primers in qPCR analysis for each group.**
(PDF)

**S4 Table. Standard curve parameters: Cycle number, amplification efficiency and limit of quantification for each assay.**
(PDF)

**S5 Table. Amplification conditions for qPCR assays.**
(PDF)

**S6 Table. Primary data on *hgcA* copy number obtained in the qPCR assays.**
(XLSX)

**S1 Fig. Monitoring stations in the northern Baltic Proper, Åland Sea and Bothnian Sea used for zooplankton collections.**
(PDF)

**S2 Fig. Standard curves obtained with the templates for three clades carrying the *hgcA* gene.**
(PDF)

**S3 Fig. Representative amplification plots and melt curves for three clades carrying the *hgcA* gene.**
(PDF)

# Acknowledgments

Crews of R/V *Aranda* (Finnish Environment Institute, SYKE, Finland), R/V *Fyrbryggaren*, support from Swedish National Marine Monitoring Programme (SNMMP) and personnel of the Askö Field Station are thanked for assisting with collection of zooplankton.

# Author Contributions

**Conceptualization:** Elena Gorokhova.

**Data curation:** Elena Gorokhova.

**Formal analysis:** Elena Gorokhova.

**Investigation:** Anne L. Soerensen.

**Methodology:** Nisha H. Motwani.

**Writing – original draft:** Elena Gorokhova.

**Writing – review & editing:** Anne L. Soerensen, Nisha H. Motwani.

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
