## [Decision Letter · Decision Letter 0]

23 Dec 2019

PONE-D-19-31877

Mercury-methylating bacteria are associated with copepods: a proof-of-principle survey in the Baltic Sea

PLOS ONE

Dear Professor Gorokhova,

Thank you for submitting your manuscript to PLOS ONE. After careful consideration, we feel that it has merit but does not fully meet PLOS ONE’s publication criteria as it currently stands. Therefore, we invite you to submit a revised version of the manuscript that addresses the points raised during the review process.

We would appreciate receiving your revised manuscript by Feb 04 2020 11:59PM. To enhance the reproducibility of your results, we recommend that if applicable you deposit your laboratory protocols in protocols.io, where a protocol can be assigned its own identifier (DOI) such that it can be cited independently in the future. For instructions see: http://journals.plos.org/plosone/s/submission-guidelines#loc-laboratory-protocols

We look forward to receiving your revised manuscript.

Kind regards,

Alberto Amato

Academic Editor

PLOS ONE

Journal Requirements:

2. Thank you for including your ethics statement:  "Stockholm University, SNMMP"

a. Please amend your current ethics statement to confirm that your named institutional review board or ethics committee specifically approved this study.

'Financial support for these project was provided by Swedish Agency for Marine and Water Management, Swedish National Monitoring Programme, and the Swedish Research Council for Environment, Agricultural Sciences and Spatial Planning (FORMAS). There is no conflict of interest, and funders had no role in governing the research or data usage.'

'The author(s) received no specific funding for this work.'

Please provide an amended Funding Statement that declares *all* the funding or sources of support received during this specific study (whether external or internal to your organization) as detailed online in our guide for authors at http://journals.plos.org/plosone/s/submit-now Please state what role the funders took in the study.  If any authors received a salary from any of your funders, please state which authors and which funder. If the funders had no role, please state: "The funders had no role in study design, data collection and analysis, decision to publish, or preparation of the manuscript."

5. We note that Figure S1 in your submission contain map images which may be copyrighted.

a. You may seek permission from the original copyright holder of Figure S1 to publish the content specifically under the CC BY 4.0 license. 

Reviewers' comments:

Reviewer's Responses to Questions

**Comments to the Author**

1. Is the manuscript technically sound, and do the data support the conclusions?

Reviewer #1: Yes

Reviewer #2: Yes

2. Has the statistical analysis been performed appropriately and rigorously? 

Reviewer #1: Yes

Reviewer #2: Yes

3. Have the authors made all data underlying the findings in their manuscript fully available?

Reviewer #1: Yes

Reviewer #2: Yes

4. Is the manuscript presented in an intelligible fashion and written in standard English?

Reviewer #1: Yes

Reviewer #2: Yes

5. Review Comments to the Author

Reviewer #1: This is a well written and concise manuscript describing the potential for mercury-methylation within zooplankton. This is potentially an important mechanism for understanding the overall mercury dynamics in aquatic food webs. The material is a bit limited making far-reaching conclusions difficult to draw. A more comprehensive study would have been achieved if authors would have waited with publication until more quantitative data was available. But they are clear about the limitations in the manuscript, and if the Editor agrees, this study could be published as a proof-of-concept study.

Apart from this general comment I only have a few minor comments.

Line 89: Year seems to be missing in the reference.

Line 128: Please give rationale for selecting these clades.

Line 241: This section is speculative. Please make sure that you refer to this as a suggestion/hypothesis that needs further testing.

End of review

Reviewer #2: Review of the manuscript PONE-D-19-31877 Mercury-methylating bacteria are associated with copepods: a proof-of-principle survey in the Baltic Sea, by Elena Gorokhova, Anne L. Soerensen and Nisha H. Motwani.

The manuscript deals with a very intriguing topic, that Hg-methylating microorganism associated with zooplankton gut microbiota might play a role in food-web transferring of MeHg. I found the topic very interesting from an ecological and ecotoxicology point of view. The manuscript is clear and well written and the experimental sections are robust. I only have some minor comments.

EXPERIMENTAL

Line 75: ‘Zooplankton were collected…….’

Please provide the experimental details for plankton sampling and sorting.

Line 110: ‘Reference samples used as a contamination control were hatched Artemia spp. ......’

Where Artemia used to test whether bacterial contamination occurred during the experimental procedure?

qPCR assay

Line 134: ……..1.5×106 to 1.5×102 apparent copies of target DNA per reaction.

Why the DNA concentration is expressed as apparent copies of target DNA and not as ng/μl? Did the authors loaded a known copy number of synthetic DNA oligonucleotides per reaction?

Discussion

As the authors discussed the importance of fecal pellets for Hg methylation, I was wondering whether there was any visible fecal material in the copepod gut during feeding on Rhodomonas salina and if these could have affected the results.

6. PLOS authors have the option to publish the peer review history of their article (what does this mean?). If published, this will include your full peer review and any attached files.

Reviewer #1: No

Reviewer #2: No

---

## [Decision Letter · Decision Letter 1]

17 Jan 2020

PONE-D-19-31877R1

Mercury-methylating bacteria are associated with copepods: a proof-of-principle survey in the Baltic Sea

PLOS ONE

Dear Professor Gorokhova,

Thank you for submitting your manuscript to PLOS ONE. After careful consideration, we feel that it has merit but does not fully meet PLOS ONE’s publication criteria as it currently stands. Therefore, we invite you to submit a revised version of the manuscript that addresses the points raised during the review process.

We would appreciate receiving your revised manuscript by Mar 02 2020 11:59PM. To enhance the reproducibility of your results, we recommend that if applicable you deposit your laboratory protocols in protocols.io, where a protocol can be assigned its own identifier (DOI) such that it can be cited independently in the future. For instructions see: http://journals.plos.org/plosone/s/submission-guidelines#loc-laboratory-protocols

We look forward to receiving your revised manuscript.

Kind regards,

Alberto Amato

Academic Editor

PLOS ONE

Additional Editor Comments (if provided):

Dear Authors,

please consider the comments from Reviwier #3 who was added to the process at this stage for logistic issues.

Thank you

Reviewers' comments:

Reviewer's Responses to Questions

**Comments to the Author**

1. If the authors have adequately addressed your comments raised in a previous round of review and you feel that this manuscript is now acceptable for publication, you may indicate that here to bypass the “Comments to the Author” section, enter your conflict of interest statement in the “Confidential to Editor” section, and submit your "Accept" recommendation.

Reviewer #1: All comments have been addressed

Reviewer #3: (No Response)

2. Is the manuscript technically sound, and do the data support the conclusions?

Reviewer #1: Yes

Reviewer #3: Partly

3. Has the statistical analysis been performed appropriately and rigorously? 

Reviewer #1: Yes

Reviewer #3: N/A

4. Have the authors made all data underlying the findings in their manuscript fully available?

Reviewer #1: Yes

Reviewer #3: No

5. Is the manuscript presented in an intelligible fashion and written in standard English?

Reviewer #1: Yes

Reviewer #3: Yes

6. Review Comments to the Author

Reviewer #1: (No Response)

Reviewer #3: General comments

The authors present an interesting story about potential for mercury-methylating bacteria in copepods, including a small sample set analyzed with previously published primers. The authors go on to speculate what the data might mean in the context of mercury distributions in the Baltic Sea.

The story is interesting, but my main concern is the specificity of the qPCR assays. What evidence do the authors have that these primers don’t amplify genomic DNA from copepods, or from some other microbial targets? Was the zooplankton food (Rhodomonas) tested for these targets?

Were the actual data provided with the manuscript (raw data showing data for all replicates etc.) I did not see them.

The text has a few issues with grammar and style. I have indicated some examples of these issues in my review below.

Specific comments

L11 food-webs >food webs

L12-14 Sentence should be reworded. It seems you refer to the sediments and detritus as a ‘source’, but it is unclear what product originates from ’the source’; the wording should be revised for clarity.

L15 Baltic > Baltic Sea

L38 in a some > in some

L43 I suggest deleting ‘thus’ here, as the sentence seems independent from the prior sentence.

L64 Baltic>Baltic Sea

L95 Was the Rhodomonas culture axenic? Did you test the presence of MeHg genes in the culture?

L139 The authors heavily cite previous studies on the qPCR method. Some more detail on qPCR approach should be provided. State that the assay was a SYBR assay and explain how you determined whether there was non-target amplification (i.e. explain results from melt curves). The DNA extraction method seems fairly ‘dirty’ as it does not include Proteinase nor a purification step beyond the Chelex step. Did you conduct inhibition tests with the copepod DNA (see my comment on the supplementary text)?

L156-157 Here you assume that there is one gene copy per cell and one genome per cell. Some justification for this is needed. I would avoid speculating this far with qPCR data, especially with all the uncertainties highlighted by the authors.

L159 Detection limit of 10^6 is alarmingly high for one of the assays (essentially means a non-detect as such amplification could easily occur with non-targets). The test efficiency of 60% is also alarmingly low, suggesting poor match with target and primers, or degraded standards or primers. The authors may consider changing the names for the primers or specifying what they actually amplify.

L166 were tested > tested

L171-173 Sentence should be reworded for clarity. The verb is very far from subject in the sentence which makes it very difficult to follow. I also don’t follow what you mean by ‘indicative’. Indicative of presence of mercury methylators? Please reword for clarity.

L176 lack of amplification of archaeal Hg-methylators?

L176-177 The sentence should be reworded for accuracy. There wasn’t a difference in amplification efficiency among bacteria - the difference was among the qPCR assays targeting different bacterial Hg-methylation genes.

Fig 1. Caption

Baltic>Baltic Sea

L185 You state that n=3-4; however, each sample was pooling many guts, correct? It would be helpful to remind the reader about it here.

L187 and 191 delete: ‘positive’ here and elsewhere where it appears with ‘amplification’

L192-193 The figure x-axis explanation is confusing. It sounds like you are saying that absence of a species on the x-axis means there were no individuals of that species collected at that station; thus, it would follow that there were no zeros (no undetects) among the samples. Are you really meaning to say that in each case you ran qPCR with copepod DNA you always detected the targets? Please clarify.

L220-221 The sentence is confusing. I suggest breaking it into two to separately state what you mean by copy number differences and how this links with phylogenetic differences. Do you mean phylogenetic differences of the microbiomes or the zooplankton?

L222 yielded

Supplementary documents

S2 Figure

Include the amplification curve equations in the figure.

S4 Table

Write out in full the names of target organisms. Include the strain identifiers.

S1 text

QPCR ‘setup’

The citations are inconsistent. At least in one case, a citation is included as a superscript, not in parentheses.

You state that you added 6 uL of DNA onto the wells, then ‘dried’ them for 30 min. This is an unusual step and should be explained.

You discuss melt curves here – explain what they showed and how it influenced what data were included. If there were several peaks the data can't be used.

Spike test

Here you explain a spike test conducted with the cladoceran DNA. Did you conduct a similar test with the copepod DNA to confirm there was no inhibition?

7. PLOS authors have the option to publish the peer review history of their article (what does this mean?). If published, this will include your full peer review and any attached files.

Reviewer #1: No

Reviewer #3: No

---

## [Editor Report · Decision Letter 2]

27 Feb 2020

Mercury-methylating bacteria are associated with copepods: a proof-of-principle survey in the Baltic Sea

PONE-D-19-31877R2

Dear Dr. Gorokhova,

We are pleased to inform you that your manuscript has been judged scientifically suitable for publication and will be formally accepted for publication once it complies with all outstanding technical requirements.

With kind regards,

Alberto Amato

Academic Editor

PLOS ONE

Additional Editor Comments (optional):

All the points raised by the Reviewer(s) have been addressed.

please notice that PLoS ONE does not edit the text hence the authors have to take care of the formatting step. I realised that in the R2 version, the doi numbers in the reference list are typed in a different character, would you pleaase harmonise the list?
---

## [Editor Report · Acceptance letter]

3 Mar 2020

PONE-D-19-31877R2 

Mercury-methylating bacteria are associated with copepods: a proof-of-principle survey in the Baltic Sea 

Dear Dr. Gorokhova:

I am pleased to inform you that your manuscript has been deemed suitable for publication in PLOS ONE. Congratulations! Your manuscript is now with our production department. 

With kind regards,

on behalf of

Dr. Alberto Amato 

Academic Editor

PLOS ONE